# SnapMem: Snapshot-based 3D Scene Memory for Embodied Exploration and Reasoning

## Abstract

Constructing a compact and informative 3D scene representation is essential for effective embodied reasoning and exploration, especially in complex environments over long periods. Existing approaches have relied on object-centric graph representations, which oversimplify 3D scenes by modeling them as individual objects and describing inter-object relationships through rigid textual descriptions. This rigidity leads to the loss of rich spatial relationships between objects, which are essential for embodied scene reasoning tasks. Furthermore, these representations lack natural mechanisms for active exploration and memory management, which hampers their applications for lifelong autonomy. In this work, we propose SnapMem, a novel 3D scene representation that leverages a compact set of informative snapshot images to cover the scene based on object co-visibility. These snapshot images capture rich spatial and semantic information among objects within the same view and their surroundings. We then illustrate how such a representation can be directly integrated with frontier-based exploration algorithms to facilitate active exploration by leveraging unexplored regions and scene memory. To support lifelong memory in active exploration settings, we further present an efficient memory aggregation pipeline to incrementally construct SnapMem, as well as an effective memory retrieval technique for memory management. Experimental results over three benchmarks demonstrate that SnapMem significantly enhances agents' reasoning and exploration capabilities in 3D environments over extended periods, highlighting its potential for advancing applications in embodied AI.

## 1 Introduction

Embodied agents operating in complex 3D environments require robust scene representations to effectively reason and explore over extended periods. Directly representing scenes using dense 3D representations, such as point clouds (Ding et al., 2023; Zhang et al., 2023; Ding et al., 2024; Jataval-labhula et al., 2023) or neural fields (Tsagkas et al., 2023; Kerr et al., 2023; Mazur et al., 2023), is often extremely computationally expensive and difficult to reason over. As a result, recent advancements have focused on object-centric representations, particularly 3D scene graphs (Wald et al., 2020; Gu et al., 2024), as a means of encoding scene memory compactly. These graphs represent scenes using nodes for objects and edges for inter-object relationships, facilitating reasoning about 3D environments.

However, existing object-centric representations exhibit significant limitations. Such representations are limited to captions or visual features in object-level, lacking flexible information at different scales. The relationships between objects, represented as edges between nodes, oversimplify the complex spatial relationships present in 3D environments. The oversimplified nature of such scene representations lacks the robustness needed for an agent to interpret intricate spatial layouts and respond to complex queries that require a nuanced understanding of both spatial and semantic information.

Moreover, these representations lack mechanisms for active exploration and effective memory management, which are essential to lifelong autonomy. In particular, agents are often deployed in partially mapped environments, and it is important that the agent has a well-specified way to explore and solve tasks. Additionally, object-centric representations will continuously grow in size due to the

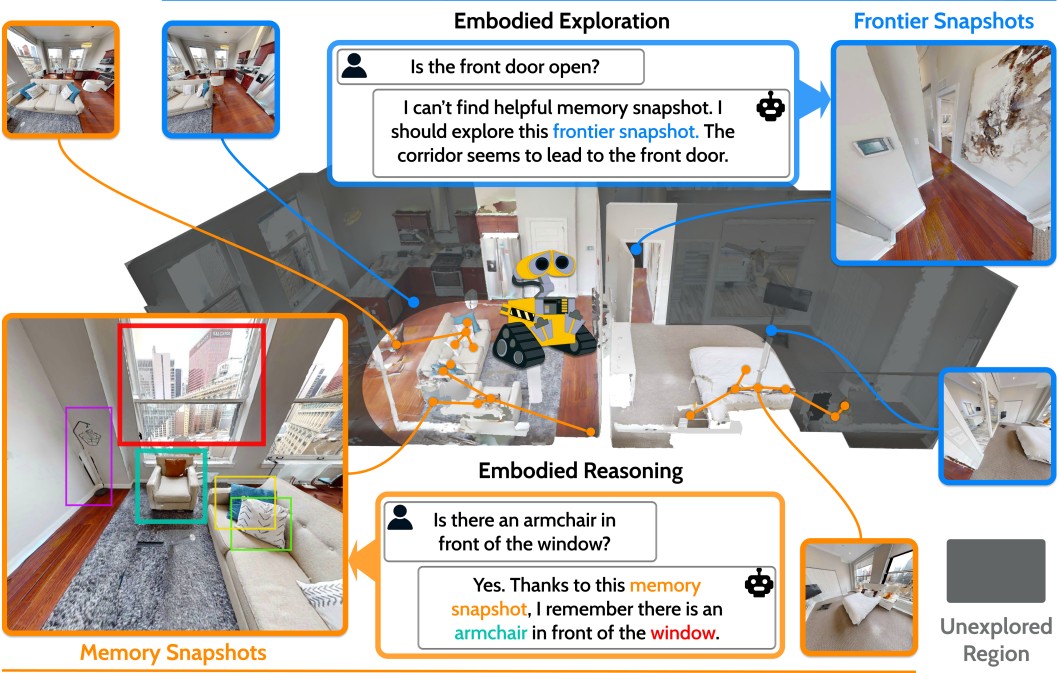

Figure 1: With SnapMem, explored regions are represented by a set of Memory Snapshots capturing clusters of co-visible objects, *i.e.*, the objects observable in a single image observation, along with their spatial relationships and background context, as shown in the bottom-left example. Unexplored regions are represented by navigable frontiers along with image observations, referred to as Frontier Snapshots. In the top-right example, the agent actively explores a frontier snapshot when no helpful memory snapshot is found.

vast number of objects in a scene, creating challenges for both storage and retrieval, and hindering long-term autonomous execution in an environment.

To address these challenges, we introduce SnapMem, a novel snapshot-based 3D scene representation that is both compact and informative. SnapMem is based on the intuition that an image alone is sufficient to capture rich visual information of a region within a 3D scene. In an image, object features and spatial relationships are directly visible, while global information can also be inferred from the background context. Therefore, SnapMem adopts a set of informative images, referred to as "memory snapshots", to represent the explored regions of a scene. These snapshots encapsulate intricate spatial and semantic information among co-visible objects and their surroundings, including background context. As illustrated in Figure 1, the memory snapshot in the bottom-left corner clearly depicts the spatial relationships among a cluster of co-visible objects, each highlighted by bounding boxes. By capturing the scene from various viewpoints, SnapMem provides richer and more robust visual information, surpassing the capabilities of traditional object-centric graphs.

In addition, SnapMem supports active exploration through integration with frontier-based exploration frameworks (Yamauchi, 1997; Mobarhani et al., 2011). As illustrated in Figure 1, we extend the concept of "frontier", which represents an unexplored region, to "frontier snapshot". Similar to a memory snapshot, we use an image observation towards the unexplored region to represent the corresponding frontier. By maintaining the frontier snapshots, the agent can make exploration decisions by leveraging both its accumulated knowledge and the potential for new information. This mechanism addresses a critical aspect of embodied reasoning by enabling the agent to actively expand its knowledge of the environment. Moreover, by representing both explored and unexplored regions with snapshot images in a unified manner, we can better leverage vision-language models (VLMs). With recent advances in VLMs' perception capabilities, these snapshots are well suited as effective inputs for visual information.

By incorporating our real-time memory aggregation and filtering framework, SnapMem serves as an effective memory system for lifelong agents operating in 3D environments. Throughout the explo-

ration process, the scene memory is dynamically and incrementally constructed, enabling agents to continuously update and refine their understanding of the environment. With each memory snapshot representing multiple objects, the size of SnapMem does not grow as large as object-centric representations during exploration. Moreover, we propose Prefiltering, a memory retrieval mechanism, that first retrieves only the relevant memory snapshots of a given query, and uses only the filtered snapshots for reasoning and planning. This allows the agent to perform continuous exploration and navigation over long periods without excessive computational burden. Extensive experiments and superior performance on three benchmarks demonstrate that SnapMem significantly enhances agents' capabilities in reasoning and lifelong exploration in 3D environments.

Our contributions can be summarized as follows:

- We introduce SnapMem, a compact scene memory that constructs informative snapshot images to capture diverse and robust information among co-visible objects and their surroundings in 3D scenes.
- By augmenting snapshot memory to include unexplored regions through frontier snapshots, we enable agents to actively explore and acquire new information. This enhancement significantly improves their abilities to complete tasks that require knowledge beyond their initial observations.
- We present a dynamic framework for SnapMem, featuring memory aggregation and filtering strategies that enable agents to actively expand their knowledge and adapt over extended periods, supporting lifelong learning in 3D environments.

## 2 RELATED WORKS

**3D Scene Representations** Recent works (Peng et al., 2023; Shafiullah et al., 2022) have focused on establishing universal 3D representations by grounding 2D representations captured by VLMs to 3D scenes, which showcases impressive results on a wide range of tasks, including navigation (Wani et al., 2020), language-guided object grounding (Hong et al., 2022). However, such representations are rather limited due to high resource consumption and the inability to support dynamic updates. 3D scene graphs address these limitations by formulating the scene as a compact graph, where nodes represent objects, and edges encode inter-object relationships as textual descriptions (Fisher et al., 2011; Gay et al., 2019; Armeni et al., 2019; Kim et al., 2019), enabling real-time establishment and dynamic update for hierarchical scene representations (Rosinol et al., 2021; Wu et al., 2021; Hughes et al., 2022). While such object-centric representations have demonstrated effectiveness in various tasks, they remain constrained for oversimplifying inter-object relationships with rigid descriptions and missing mechanism for active exploration and memory management. To tackle this challenge, our work leverages a set of informative snapshot images to visually capture spatial and semantic relationships among objects, offering a more sophisticated understanding of the scene.

**VLM for Exploration and Reasoning** Vision-Language Models (VLMs) have shown promising results in solving embodied exploration and reasoning tasks by leveraging commonsense reasoning and internet-scale knowledge. Existing exploration approaches can be divided into two categories. The former directly employs consecutive observations together with instructions as input, requiring the VLM to predict next-step action (Zhang et al., 2024) while the latter grounds the exploration target to 3D scene through visual prompting, establishing a semantic map to guide the exploration process (Majumdar et al., 2022; Shah et al., 2023; Ren et al., 2024; Yokoyama et al., 2024). However, both approaches are constrained by their memory representations. For the former, vanilla past observations can only serve as short-term memory. For the latter, their semantic maps are target-specific and cannot be generalized to future tasks. On the other hand, current reasoning approaches generally assume a fully observable scene as the input of the VLM, either represented with image observations (Chen et al., 2024) or 3D scene representations like point clouds (Hong et al., 2023), which makes them inapplicable in partially mapped environments. To address these limitations, our work introduces the first lifelong and target-agnostic scene memory that can be seamlessly integrated with VLM for further reasoning, stepping closer to the ultimate goal of lifelong autonomy.

## 3 APPROACH

In this section, we first introduce how SnapMem is constructed from a series of RGB-D images with poses using co-visibility clustering (Section 3.1). We then explain how SnapMem can be integrated

with frontier-based exploration and incrementally and dynamically constructed during exploration (Section 3.2).

---

**Algorithm 1** Co-Visibility Clustering for Memory Snapshots

---

1: Initial clusters $\mathcal{C} = \{\mathcal{O}\}$
2: Temporary memory snapshot set $\mathcal{S}_{tmp} = \varnothing$, Final memory snapshot set $\mathcal{S} = \varnothing$
3: All frame candidates $\mathcal{I}$
4: Score function $\mathcal{F}$
5: **while** $\mathcal{C}$ is not empty **do**
6:     $\mathcal{O}^* = \arg\max_{\mathcal{O} \in \mathcal{C}} \|\mathcal{O}\|$
7:     $\mathcal{I}^* = \{I | I \in \mathcal{I}, \mathcal{O}^* \subseteq \mathcal{O}_I\}$
8:     **if** $\mathcal{I}^*$ is not empty **then**
9:         $I^* = \arg\max_{I \in \mathcal{I}^*} \mathcal{F}(I)$
10:        $S^* = <\mathcal{O}^*, I^*>$
11:        $\mathcal{S}_{tmp} = \mathcal{S}_{tmp} \cup \{S^*\}$
12:     **else**
13:        Use K-Means to split $\mathcal{O}^*$ into two clusters $\mathcal{O}^* = \mathcal{O}_1^* \cup \mathcal{O}_2^*$ based on $(x, y, z)$ coordinates
14:        $\mathcal{C} = \mathcal{C} \cup \{\mathcal{O}_1^*, \mathcal{O}_2^*\}$
15:     **end if**
16:     $\mathcal{C} = \mathcal{C} - \{\mathcal{O}^*\}$
17: **end while**
18: $\mathcal{I}_{\mathcal{S}} = \{I_S | S \in \mathcal{S}_{tmp}\}$
19: **for** $I \in \mathcal{I}_{\mathcal{S}}$ **do**
20:     $S = <\cup_{S \in \mathcal{S}_{tmp}, I_S = I} \mathcal{O}_S, I>$
21:     $\mathcal{S} = \mathcal{S} \cup \{S_m\}$
22: **end for**
    return $\mathcal{S}$

---

## 3.1 SnapMem Construction

Inspired by the idea that an image itself is informative enough to represent a small area of the scene with rich and robust information, we propose a novel way to utilize a set of snapshot images to cover the whole informative areas of a scene. Instead of the object-centric representation proposed by ConceptGraph, in which only object-level visual features are stored and managed, we propose using one image to represent a cluster of objects that are co-visible in that image, namely a **Memory Snapshot**. With this, the major objects in a scene can be visually represented by a small set of images.

Specifically, given a set of $N$ image observations $\mathcal{I}^{obs} = \{I_1^{obs}, I_2^{obs}, ..., I_N^{obs}\}$, where each $I_i^{obs} = \langle I_i^{rgb}, I_i^{depth}, \theta_i \rangle$ (color image, depth, pose), we first utilize ConceptGraph (Gu et al., 2024) pipeline to do a series of object detection, segmentation, spatial transformations and merging, resulting in an object set that contains all detected objects from the observations $\mathcal{O} = \{o_1, o_2, ..., o_M\}$ of size $M$, where each object $o_j = \langle c_j, p_j \rangle$ is characterized by an object category and its 3D location. Meanwhile, we obtain a set of frame candidates $\mathcal{I} = \{I_1, I_2, ..., I_N\}$, where each $I_i = \langle I_i^{obs}, \mathcal{O}_{I_i} \rangle$ consists of the image observation together with a list of all detected objects in that image, *i.e.*, all objects in $\mathcal{O}_{I_i}$ are co-visible in $I_i^{obs}$.

We define SnapMem $\mathcal{S}$ as a set of memory snapshots $\{S_1, S_2, ..., S_K\}$ of size $K \leq N$, where each memory snapshot $S_k = \langle \mathcal{O}_{S_k}, I_{S_k} \rangle$ is characterized by a frame candidate $I_{S_k} \in \mathcal{I}$ and a cluster of objects $\mathcal{O}_{S_k}$ that is a subset of all detected objects in the image $I_{S_k}^{obs}$, *i.e.*, $\mathcal{O}_{S_k} \subseteq \mathcal{O}_{I_{S_k}}$. Therefore, an image $I_{S_k}^{obs}$ serves as a shared visual feature of the group of objects $\mathcal{O}_{S_k}$. Since $\mathcal{S}$ needs to cover the whole object set $\mathcal{O}$, and each object $o_j$ needs to be uniquely represented by one memory snapshot $S_k$ (although it may still be visible in other snapshot images), we require $\mathcal{O}_{S_1} \cup \mathcal{O}_{S_2} \cup ... \cup \mathcal{O}_{S_K} = \mathcal{O}$, and $\mathcal{O}_{S_i} \cap \mathcal{O}_{S_j} = \varnothing$ for $\forall S_i, S_j \in \mathcal{S}$.

To acquire the desired set of memory snapshots, we follow Savaresi & Boley (2001) to hierarchically split $\mathcal{O}$ into clusters, each of which is a subset of the detected object list $\mathcal{O}_{I_i}$ of a certain frame candidate $I_i$. As detailed in the pseudocode in Algorithm 1, we define a cluster set $\mathcal{C}$ composed of all unsettled object clusters that haven't been matched with observations, initialized to contain the full object set $\{\mathcal{O}\}$, and the temporary memory snapshot set $\mathcal{S}_{tmp}$, initialized to $\varnothing$. Each time, we

pick the largest unsettled cluster $\mathcal{O}^*$ from $\mathcal{C}$ and search through all frame candidates for capable candidates $I^*$ such that $\mathcal{O}^*$ is a subset of the detected object list of $I^*$. When such candidates exist, we rank them based on a score function $\mathcal{F}$ and pick the top-ranked frame candidate $I^*$ to create a new memory snapshot $S^* = < \mathcal{O}^*, I^* >$ and add it to $\mathcal{S}_{tmp}$. In practice, we choose $\mathcal{F}(I) = \|\mathcal{O}_I\|$ to select the observation that not only covers most objects but also has the highest sum of confidence for all objects in the cluster. If there is no feasible frame candidate, we then use K-Means to further divide $\mathcal{O}^*$ into two subclusters $\mathcal{O}_1^*$ and $\mathcal{O}_2^*$ based on the 2D horizontal positions of the objects, and add them to $\mathcal{C}$. We repeat the above process until no clusters remain in $\mathcal{C}$. Note that the process is guaranteed to terminate for every object that has been captured in certain observations. Ultimately, after all objects have been assigned to corresponding snapshots, we merge memory snapshots in $\mathcal{S}_{tmp}$ that share the same observations, achieving the final compact memory representation $\mathcal{S}$.

In each memory snapshot, not only is the visual information of each object stored, but also the spatial relationships between objects and the room-level information are provided by visual cues in the background. With the increasing perception abilities of VLMs, such snapshot-based representations can provide richer and more robust visual information for VLMs to complete difficult tasks.

### 3.2 SNAPMEM WITH FRONTIER-BASED DYNAMIC EXPLORATION

#### 3.2.1 INTEGRATION WITH FRONTIER-BASED EXPLORATION

We adapt the frontier-based exploration pipeline from Ren et al. (2024). In a frontier-based exploration episode, an agent is initialized in an unknown scene and explores the environment step by step. At each step, the agent moves to a new location and receives a series of observations, including depth and pose. The depth images are mapped into a 3D occupancy map, which allows us to determine which areas are navigable. Meanwhile, we record a map of the explored regions, defined as the nearby areas along the agent's trajectory, and a map of the unexplored regions, defined as navigable but yet-to-be-explored areas. A frontier is then defined to represent such an unexplored region that could be further explored.

In this work, we extend this concept by using a snapshot to represent a frontier, similar to memory snapshots. We define a **Frontier Snapshot** $F = \langle r, p, I^{obs} \rangle$, consisting of the unexplored region $r$ it represents, a navigable location $p$, and an image observation $I^{obs}$ from the agent's position toward that unexplored region. Therefore, the frontier shares the same format as memory snapshots, and both can be used jointly as inputs into VLMs. More implementation details about frontier-based exploration are in Appendix A.1.

#### 3.2.2 INCREMENTAL CONSTRUCTION OF SNAPMEM

Throughout the exploration process, the scene memory is dynamically and incrementally constructed. At each exploration step, the agent observes its surroundings and updates the scene memory and frontiers. At step $t$, we denote the current object set as $\mathcal{O}_t$, the frontier set as $\mathcal{F}_t$, the memory snapshot set as $\mathcal{S}_t$, and the frame candidate set as $\mathcal{I}_t$, all of which are initialized as $\varnothing$ at the beginning of the episode.

**Detect.** As illustrated in Figure 2, at each time step $t$, the agent first captures $N$ egocentric views $\mathcal{I}^{obs} = \{I_1^{obs}, I_2^{obs}, ..., I_N^{obs}\}$. The ConceptGraph pipeline is then applied to $\mathcal{I}^{obs}$ to extract the object set $\mathcal{O}$ and frame candidate set $\mathcal{I}$: $\mathcal{O}, \mathcal{I} = \text{ConceptGraph}(\mathcal{I}^{obs}, max\_dist)$. Specifically, the threshold "$max\_dist$" ensures that only objects within a certain distance from the agent are added to the scene graph, as the memory snapshot should only represent objects from a local area. It is important to note that the object set $\mathcal{O}$ detected in these egocentric views may contain both newly identified objects and those already present in the previous set $\mathcal{O}_{t-1}$. Subsequently, the full object set and frame candidate set are updated as $\mathcal{O}_t = \mathcal{O}_{t-1} \cup \mathcal{O}$ and $\mathcal{I}_t = \mathcal{I}_{t-1} + \mathcal{I}$ respectively.

**Cluster.** We implement the co-visibility clustering in Section 3.1 incrementally. At each time step $t$, instead of performing clustering on the entire object set $\mathcal{O}_t$, we focus on clustering objects related to $\mathcal{O}$, the objects detected from the egocentric views at this step. In $\mathcal{O}$, some objects may have already been assigned to specific memory snapshots in $\mathcal{S}_{t-1}$. We refer to those memory snapshots as $\mathcal{S}_{prev} = \{S | S \in \mathcal{S}_{t-1}, \mathcal{O}_S \cap \mathcal{O} \neq \varnothing\}$. All objects from $\mathcal{S}_{prev}$, along with the newly detected objects in $\mathcal{O}$, are used as input for clustering, denoted as $\mathcal{O}_{input}$. Then, the memory snapshot set is updated as $\mathcal{S}_t = \mathcal{S}_{t-1} - \mathcal{S}_{prev} + \text{Cluster}(\mathcal{O}_{input}, \mathcal{I}_t)$

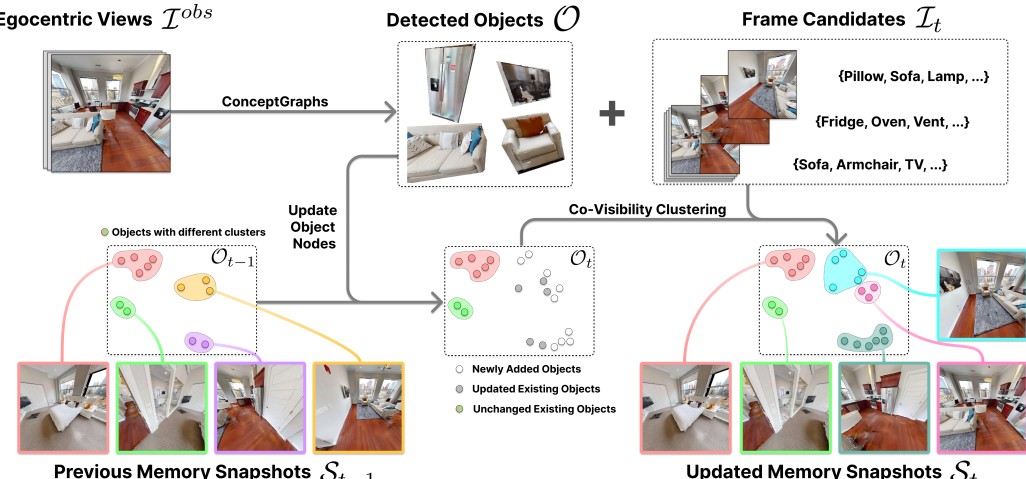

Figure 2: **The memory aggregation process of SnapMem.** At each step $t$, the object set $\mathcal{O}_t$ is first updated using the object-wise update pipeline from ConceptGraph. The newly detected objects and the updated existing objects are then jointly clustered into new memory snapshots using co-visibility clustering (Algorithm 1), which are used to update the memory snapshot set $\mathcal{S}_t$.

**Frontier Update.** At each step $t$, an existing frontier from $\mathcal{F}_{t-1}$ may be modified if the unexplored region it represents has been updated, or it may be removed if the region has been fully explored. Additionally, new frontiers may be introduced. For each newly added or modified frontier, a snapshot is taken to update its image representation. As a result, $\mathcal{F}_{t-1}$ is updated to $\mathcal{F}_t$.

More implementation details regarding how the agent moves and navigates are in Appendix A.2.

### 3.2.3 MEMORY RETRIEVAL WITH PREFILTERING

For a given instruction, most memory snapshots are irrelevant, and that processing these irrelevant snapshots consumes substantial computational resources without contributing meaningful information. Therefore, we introduce a novel memory retrieval mechanism called **Prefiltering**. Figure 3 illustrates Prefiltering in an embodied question answering task. We present the VLM with the question, along with all object categories in $\mathcal{O}_t$. The VLM is then tasked with outputting all relevant object categories in the order of relevancy and importance, and a hyperparameter $K$ is employed to keep only the top $K$ categories. Memory snapshots that do not contain any object within the selected categories are filtered out. This prefiltering technique significantly reduces resource consumption, allowing us to include images directly within the prompt. Moreover, prefiltering can help eliminate many falsely detected objects caused by the limitations of the object detection model, increasing the robustness of SnapMem. The complete prompt is provided in Appendix A.5.

### 3.2.4 REASONING AND EXPLORATION WITH VLMS

With the updated frontier snapshots and memory snapshots, we can directly leverage the perception and reasoning capabilities of large VLMs, as the snapshot-based nature of frontier and memory snapshots makes them easily interpreted by VLMs.

SnapMem is versatile and can be prompted in various ways for different tasks. In the case of embodied question answering (illustrated in Figure 3), the VLM is required to either choose a frontier to explore or answer the question based on the memory snapshots. If the VLM chooses a frontier, it must provide a rationale for exploring in that direction; otherwise, it must directly provide an answer to the question, which is then adopted as the final answer for that exploration episode. In object navigation tasks, where the agent is tasked with finding a specific object, we modify the prompt by appending each memory snapshot with the image crops of the objects it contains, and the VLM is required to directly pick an object from one memory snapshot. Detailed experiments on these two tasks are presented in Section 4.1 and 4.3 respectively, with the complete prompt provided in Appendix A.5.

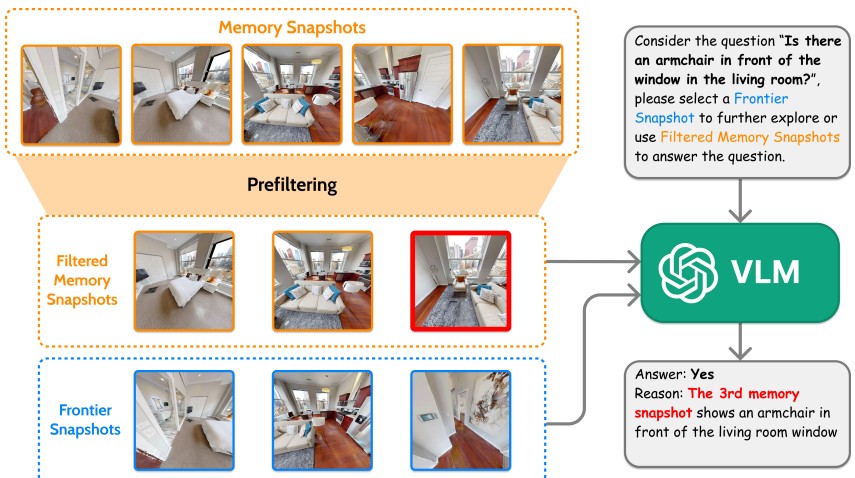

Figure 3: **SnapMem as visual input for the VLM in embodied question answering.** The VLM first retrieves relevant memory snapshots with prefiltering, then utilizes the frontier snapshots and memory snapshots to perceive the scene and reason about the embodied questions.

## 4 EXPERIMENTS

SnapMem is a form of scene representation that stores rich and compact visual information, serving as a memory system for a lifelong agent to explore and reason about a scene. To comprehensively evaluate SnapMem, we begin with Active Embodied Question Answering (Section 4.1), where the scene is initially unknown. This assessment tests SnapMem's overall performance in scenarios that require both embodied exploration and reasoning. Next, we examine SnapMem's efficiency in representing 3D scene information through Episodic Memory Embodied Question Answering (Section 4.2). In this evaluation, the scene scan of the ground truth region is provided and no exploration is needed. Following this, we evaluate SnapMem on GOAT-Bench (Section 4.3), a multi-modal lifelong navigation benchmark, to demonstrate SnapMem's effectiveness as a lifelong memory system. Finally, we conduct a series of ablation studies to determine key hyperparameters choices in Appendix A.4.

For all experiments, we construct SnapMem based on the real-time streamlined implementation of ConceptGraphs, using YOLO-World-X (Cheng et al., 2024) as our object detector. Since Snap-Mem is a versatile scene memory, we adapt it to different benchmarks in slightly different ways, as explained in each respective subsection. More implementation details are in Appendix A.2.

### 4.1 ACTIVE EMBODIED QUESTION ANSWERING (A-EQA)

On the A-EQA (Majumdar et al., 2024) benchmark (Table 1), we evaluate SnapMem's ability to dynamically construct scene representations for exploration and reasoning given complex questions.

**Benchmark.** A-EQA consists of 557 questions drawn from 63 scenes in HM3D (Ramakrishnan et al., 2021). Due to resource limitations, our evaluation focuses on a subset of 184 questions, as mentioned in the OpenEQA benchmark (Majumdar et al., 2024). The open-vocabulary and open-ended questions in A-EQA encompass diverse daily tasks such as object recognition, functional reasoning, and spatial understanding. For each question, an agent is initialized at a specific location and is required to explore the scene to gather the necessary information for answering the question.

**Implementation Details.** As explained in detail in Section 3.2, we integrate SnapMem into the frontier-based exploration framework. The VLM directly returns an answer after identifying visual clues from certain memory snapshots. We set the number of egocentric observations at each step $N = 3$, the maximum distance for objects to be included in the scene graph $max\_dist = 3.5$, and the number of prefiltered classes $K = 10$.

**Metrics.** Following OpenEQA, we employ LLM-Match and LLM-Match SPL for quantitative evaluation. We first rate each predicted answer from 1 to 5 using GPT-4 to compare ground-truth and

| Method | LLM-Match ↑ | LLM-Match SPL ↑ |
|---|---|---|
| ***Blind LLMs*** | | |
| GPT-4* | 35.5 | N/A |
| GPT-4o | 35.9 | N/A |
| ***Question Agnostic Exploration*** | | |
| CG Scene-Graph Captions* | 34.4 | 6.5 |
| SVM Scene-Graph Captions* | 34.2 | 6.4 |
| LLaVA-1.5 Frame Captions* | 38.1 | 7.0 |
| Multi-Frame* | 41.8 | 7.5 |
| ***VLM Exploration*** | | |
| Explore-EQA | 46.9 | 23.4 |
| CG w/ Frontier Snapshots | 47.2 | 33.3 |
| SnapMem (Ours) | **52.6** | **42.0** |
| Human Agent* | 85.1 | N/A |

Table 1: Experiments on A-EQA. "CG" denotes ConceptGraphs. Methods with * are reported from OpenEQA (Majumdar et al., 2024).

predicted answers. Given the predicted answers, LLM-Match, which measures the answer accuracy, is calculated as the average score for each question, mapped to a 20-100 scale. LLM-Match SPL, which measures the exploration efficiency, is then calculated by weighting the LLM-Match score by exploration path length. For the questions where the VLM Exploration methods failed to provide an answer, we ask GPT-4o to directly guess an answer without visual inputs, setting the SPL to 0.0.

**Baselines.** For baselines that use VLM for exploration, we mainly compare SnapMem with Explore-EQA (Ren et al., 2024) and ConceptGraph (Gu et al., 2024) w/ frontier snapshots. We adapt Explore-EQA for open-ended questions by halting exploration and answering the question with the ego-centric view once the VLM's confidence in the question exceeds a predetermined threshold. We integrate ConceptGraph into our exploration pipeline by replacing memory snapshots with object image crops, while maintaining other settings the same, including prefiltering and how answers are obtained. We adopt GPT-4o as the choice of VLM by directly utilizing the OpenAI API. Besides the methods that can do active exploration above, we also include other simple baselines implemented by OpenEQA. The group of question-agnostic exploration baselines employ question-agnostic frontier exploration to obtain an episodic memory of image frames. These frames are subsequently used to prompt VLMs directly (Multi-Frame), generate frame captions as prompts for LLMs (LLaVA-1.5 Frame-Captions), or construct textual scene-graph representation using ConceptGraph (CG) and Sparse Voxel Map (SVM) to prompt LLMs. Additionally, blind LLM experiments are included, where the LLM is tasked with answering questions without any visual information. Note that the Multi-Frame baseline uses 75 frames for each question, and is evaluated on the 184-question subset. Other baselines from OpenEQA are evaluated on the full 557-question set.

**Analysis.** As shown in Table 1, SnapMem significantly outperforms previous methods in both accuracy and efficiency. The superior performance in open-ended embodied question answering highlights the advantages of using snapshots as a memory format, which can store richer and more flexible visual information for the VLM to address complex questions. In contrast, object-based memory systems—using either image crops or language captions to represent objects and spatial relationships—are less robust when handling diverse questions, as they rely on rigid object-level features. Additionally, the multi-frame VLM implemented by OpenEQA also achieves inferior results, despite using a similar snapshot-based representation. Multi-Frame with linearly selected frames include too much repetitive or irrelevant information for the questions. This result, in turn, demonstrates the compactness and efficiency of SnapMem as a scene memory system.

## 4.2 EPISODIC-MEMORY EMBODIED QUESTION ANSWERING (EM-EQA)

We evaluate the representation capability of SnapMem on EM-EQA (Majumdar et al., 2024) to further demonstrate 1) the effectiveness of image memory compared to captions, 2) the compact and informative nature of our method.

| Methods | Avg. Frames | LLM-Match |
|---|---|---|
| Blind LLM* | 0 | 35.5 |
| CG Captions* | 0 | 34.4 |
| SVM Captions* | 0 | 34.2 |
| Frame Captions* | 0 | 38.1 |
| Multi-Frame | 3.0 | 48.1 |
| SnapMem (Ours) | 3.1 | **57.2** |
| Human | Full | 86.8 |

Table 2: EM-EQA Experiments. Frame Efficiency and performance. Methods denoted by * use GPT-4 to generate answers, as reported in OpenEQA

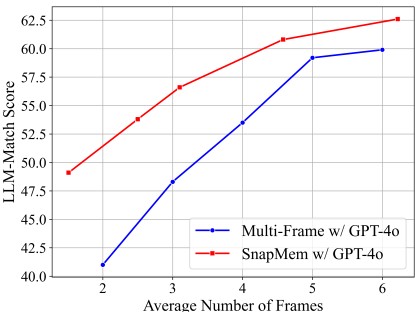

Figure 4: LLM-Match Score vs. Average Number of Frames for SnapMem and Multi-Frame both using GPT-4o

**Benchmark.** EM-EQA is an Embodied Q&A benchmark that contains over 1600 questions from 152 ScanNet (Dai et al., 2017) and HM3D(Ramakrishnan et al., 2021) scenes. The open-vocabulary and open-ended questions in EM-EQA encompass diverse daily tasks such as object recognition, functional reasoning, and spatial understanding. For each question, a trajectory comprising RGB-D observations and the corresponding camera poses at each step is provided, offering necessary contextual information needed to answer the questions.

**Implementation Details.** To adapt SnapMem to the EM-EQA benchmark, we first construct Snap-Mem for each scene using the given RGB-D observations and corresponding camera poses. For each question, we then apply prefiltering to the memory snapshots using different K values (1, 2, 3, 5, 10), and utilize the resulting filtered snapshots as prompts for GPT-4o to generate the answers.

**Baselines.** We compare against language-only scene representations, including ConceptGraphs captions, Sparse Voxel Maps Captions, and Frame Captions. We also compare against Multi-Frame, which directly processes 2 to 6 linearly sampled frames using GPT-4o.

**Analysis.** As shown in Table 2, both SnapMem and Multi-Frame significantly outperform methods that rely on captions to represent a 3D scene while using only approximately three frames. This demonstrates the effectiveness of using a set of images to represent a 3D scene and highlights the limitations of 3D scene graph captions when addressing complex queries involving relationships between objects. Furthermore, in both Table 2 and Figure 4, we observe that SnapMem surpasses Multi-Frame in frame efficiency, underscoring the compact and informative nature of our proposed 3D scene memory.

### 4.3 GOAT-BENCH

On GOAT-Bench (Khanna* et al., 2024) (Table 3), we evaluate SnapMem's effectiveness as a life-long memory system that facilitates efficient exploration and reasoning.

**Benchmark.** GOAT-Bench is a multimodal lifelong navigation benchmark, where an agent is tasked with sequentially navigating to several objects in an unknown scene, with each target described by either a category name (*e.g.*, microwave), a language description (*e.g.*, the microwave on the kitchen cabinet near the fridge), or an image of the target object. Due to the large size of GOAT-Bench and the resource limitations, we assess a subset of the "Val Unseen" split, consisting of one exploration episode for each of the 36 scenes, totaling 278 navigation subtasks.

**Implementation Details.** We reformulate the navigation task into the embodied question answering format by filling in templates for three types of target descriptions: "Can you find the {category}?", "Can you find the object described as {language description}?", and "Can you find the object captured in the following image? {image}". We adapt the prompt for navigation tasks as described in Section 3.2.4, allowing the VLM to choose an object directly from a memory snapshot. After the VLM identifies an object in such a way, the agent navigates to a location near that object to complete the task. We evaluate both GPT-4o and open-sourced VLM (specifically LLaVA-7B (Liu et al., 2023)) as the choice of VLM. For LLaVA-7B model, we further fine-tune it on our generated dataset for better performance (see Appendix A.3 for more details). Other hyperparameter settings are the same as the experiments on A-EQA.

**Metrics.** GOAT-Bench employs the Success Rate and Success weighted by Path Length (SPL) as metrics, similar to A-EQA dataset. A navigation task is deemed success if the agent's final location is within 1 meter from the navigation goal. SPL is the success score weighted by exploration distances.

**Baselines.** Similar to the experiments in A-EQA, we compare SnapMem with Explore-EQA (Ren et al., 2024) and ConceptGraph (Gu et al., 2024) baselines. Due to implementation differences in Explore-EQA, we introduce an additional success criterion for this baseline: a subtask is considered successful if the target object is visible in the final observation. This supplementary criterion leverages ground truth grounding, thereby enhancing the baseline's capability. To demonstrate the effectiveness of SnapMem's lifelong memory, we include another baseline (SnapMem w/o memory) in which we clear the constructed scene graph after each navigation task. We also directly include baselines implemented in GOAT-Bench. However, these baselines are simple RNN-based models trained via reinforcement learning, which causes their performance to lag behind the baselines we implemented.

**Analysis.** As shown in Table 3, SnapMem achieves the highest scores compared to previous methods in both accuracy and efficiency. Even though GOAT-Bench is an object-based navigation benchmark, which is well-suited for ConceptGraph settings, SnapMem still outperforms ConceptGraph w/ frontier snapshots. This can be attributed to the snapshot-based representation, which captures more comprehensive information, making it easier to match with the diverse descriptions in GOAT-Bench. Furthermore, when compared with the original SnapMem, the performance of SnapMem w/o memory declines for both GPT-4o and LLaVA-7B models, particularly in efficiency (SPL), indicating that SnapMem is beneficial as a memory system for lifelong learning. Additionally, Explore-EQA, which uses a traditional value map for each subtask to indicate regions of interest, also performs worse, as it lacks the mechanism to memorize information in explored regions.

| Method | Success Rate ↑ | SPL ↑ |
|---|---|---|
| ***GOAT-Bench Baselines*** | | |
| Modular GOAT* | 24.9 | 17.2 |
| Modular CLIP on Wheels* | 16.1 | 10.4 |
| SenseAct-NN Skill Chain* | 29.5 | 11.3 |
| SenseAct-NN Monolithic* | 12.3 | 6.8 |
| ***Open-Sourced VLM Exploration*** | | |
| SnapMem w/o memory | 40.6 | 14.6 |
| SnapMem (Ours) | 49.6 | 29.4 |
| ***GPT-4o Exploration*** | | |
| Explore-EQA | 55.0 | 37.9 |
| CG w/ Frontier Snapshots | 61.5 | 45.3 |
| SnapMem w/o memory | 58.6 | 38.5 |
| SnapMem (Ours) | **69.1** | **48.9** |

Table 3: Experiments on the subset of the GOAT-Bench "Val Unseen" split. "CG" denotes Concept-Graphs. Methods denoted by * are from GOAT-Bench.

## 5 CONCLUSION

We present SnapMem, a snapshot-based 3D scene memory that uses a set of informative snapshot images to cover the scene and store robust visual information. With the integration of the frontier-based exploration framework, SnapMem allows the agent to either leverage the memory of explored regions to solve tasks or explore the scene to expand its knowledge. With its incremental construction and efficient memory retrieval mechanism, SnapMem serves as an effective memory system for lifelong agents. Extensive experiments demonstrate the significant advantages of SnapMem over traditional scene representations.

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

# A APPENDIX

## A.1 DETAILS OF FRONTIER-BASED EXPLORATION FRAMEWORK

Our frontier-based exploration framework is based on the framework in Explore-EQA (Ren et al., 2024). We enhance its robustness and adapt it to our snapshot-based representation framework. A 3D grid-based occupancy map $M$, representing the length, width and height of the entire room, is used to record the occupancy, with each voxel having a side length of 0.1 meters. During exploration, each depth observation, together with its corresponding observation pose, is used to map unoccupied spaces onto the initially fully occupied $M$. The navigable region is then defined as the layer of unoccupied voxels at the height of 0.4 meters above the ground where the agent moves. Within this navigable region, the area within 1.7 meters of the agent's trajectory is defined as the explored region, while the remainder is designated as the unexplored region, as illustrated in Figure 5.

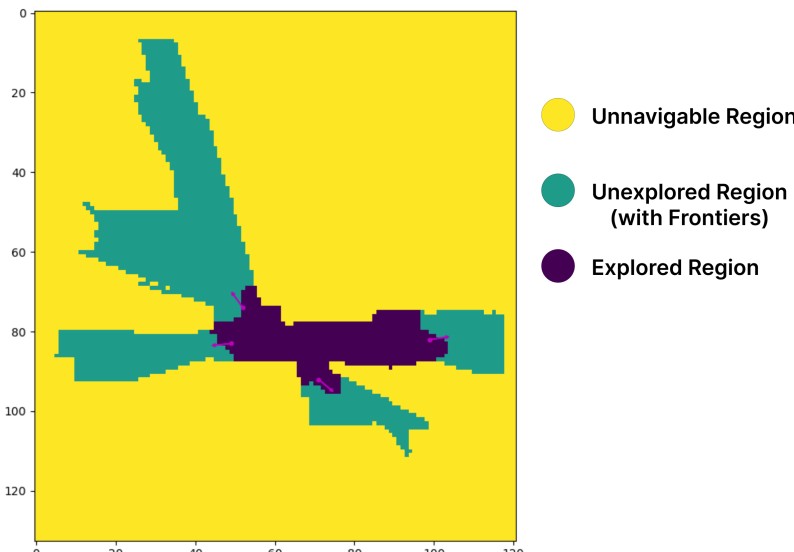

Figure 5: A illustration of different regions and frontiers in the frontier-based exploration framework. Note that navigable region consists of explored and unexplored regions.

Frontiers are defined as clusters of pixels in the unexplored region. Pixels in the unexplored region are clustered into different groups using Density-Based Spatial Clustering of Applications with Noise (DBSCAN), with each group consisting of connected pixels. Each frontier $F = \langle r, p, I^{obs} \rangle$ represents such a pixel group $r$. The navigable location of the frontier $p$ is determined at the boundary between the frontier region and the explored region, and an image observation $I^{obs}$ is captured once the frontier has been updated. As shown in Figure 5, each purple arrow together with a green region it points to is a frontier. For a frontier to be meaningful, $r$ must contain more than 20 pixels; otherwise, the frontier will not be created. A frontier is considered updated if the intersection-over-union (IoU) between the new and previous regions $r$ is less than 0.95. Additionally, if $r$ spans more than $150°$ in the agent's field of view, it is split into two regions using K-Means clustering, resulting in two separate frontiers. This approach allows for more flexibility in choosing navigation directions. Also, it is important to note that this format for representing 3D space does not currently support scenes with multiple floors. Consequently, our results in Table 1 fall significantly short of human performance, as many of the questions in A-EQA require exploration across different floors.

When prompting the VLM, only the image observations are included in the prompt. If the VLM chooses a frontier $F$, the location $p$ is used as the agent's navigation target.

## A.2 DETAILS OF THE ACTIVE EXPLORATION FRAMEWORK

At each step $t$, we take $N = 3$ egocentric views, each with a gap of $60°$. The egocentric views are captured at a resolution of $1280 \times 1280$ for better object detection and are then resized to $360 \times 360$

as frame candidates for VLM input. Frontier snapshots are initially captured at $360 \times 360$. We use YOLOv8x-World, implemented by Ultralytics, as our detection model and a 200-class set from ScanNet (Dai et al., 2017) as the detection class set. Then, we provide the VLM with the filtered memory snapshots, frontier snapshots, and an egocentric view in the forward direction.

When prompting the VLM for embodied question answering (A-EQA Benchmark), as shown in Figure 10, we append each memory snapshot with the object classes it contains. However, we only append classes that are within the prefiltered class list. The VLM will then respond with either a frontier snapshot or a memory snapshot. If the VLM returns a frontier, we set the location $p$ as the navigation target. If the VLM returns a memory snapshot along with the answer, although we directly conclude the navigation episode in our A-EQA experiments, we also set a navigation target for that memory snapshot. This allows the agent to move closer to the snapshot region, refine the selected memory snapshot, and potentially reconsider its choice.

The navigation location for a memory snapshot is determined by several conditions. We set the observation distance, $obs\_dist$, to 0.75 meters. If the snapshot contains only one object, the location is set $obs\_dist$ away from the object, in the direction from the object's location toward the center of the navigable area that is $obs\_dist$ around the object. If the memory snapshot contains two objects, the location is set $obs\_dist$ away from the midpoint of the two objects, in the direction of the perpendicular bisector of the line segment connecting the objects. If the memory snapshot contains more than two objects, we first perform Principal Component Analysis (PCA) on the object cluster to obtain the principal axis with the smallest eigenvalue. The navigation location is then set $obs\_dist$ away from the center of the object cluster, in the direction of this principal axis. Note that, in all cases for determining the navigation location, we always ignore the height of the objects and treat them as 2D points. Additionally, the above algorithm can be randomized by assigning the highest probabilities to the aforementioned positions.

Embodied navigation tasks (GOAT-Bench Benchmark) work similarly, with the following differences: 1) we append the object crop after each class name when prompting the VLM, as shown in the prompt in Figure 11; 2) when the VLM returns an object choice, we treat that object as a memory snapshot containing one object and follow a similar method to set the navigation location.

After a navigation target is set (either a frontier or a memory snapshot), the agent moves 1 meter along a path generated by the pathfinder in habitat-sim (Savva et al., 2019; Szot et al., 2021; Puig et al., 2023). Although we utilize the pathfinder, which uses prior information from a global navmesh to find the shortest paths, we can easily replace it with a simple path-finding algorithm based on the navigable map described in Appendix A.1. Step $t$ ends after the movement. Then in the new step $t + 1$, the agent updates the frontiers and memory snapshots and makes the next decision. We set a maximum of 50 steps for each navigation task.

### A.3 DETAILS OF TRAINING OPEN-SOURCED VLMS FOR GOAT-BENCH NAVIGATION

#### A.3.1 TRAINING DATASET COLLECTION

In GOAT-Bench (Khanna* et al., 2024), each navigation target is described by three types of descriptors: category, language, and image. We generate training data based on their provided exploration data, sourced from 136 scenes in HM3D (Ramakrishnan et al., 2021) training set. In each scene, a set of navigation targets is provided, each consisting of an object ID, location, category, language description, and multiple viewpoints and angles for capturing image observations. In total, the training set includes 3669 such objects, which we use as navigation targets to generate training data in our framework's format.

We adapt our exploration pipeline for data generation. For each navigation target, we first randomly select an initial point on the same floor. We then use the pathfinder in habitat-sim (Savva et al., 2019; Szot et al., 2021; Puig et al., 2023) to find the shortest trajectory to the target. At each step, if the target object is present in a memory snapshot, we use that memory snapshot as the ground truth and move one step toward a location near it; if the target object is not present in any memory snapshot, we select the frontier closest to the shortest trajectory as the ground truth for that step and move one step toward that frontier. On average, we collect 4 exploration paths per target object from different initial points, with each path consisting of approximately 12 steps.

We also collect the ground truth for prefiltering by prompting GPT-4o. For each navigation target, we collect all objects that can be seen along the exploration path and feed them, together with the description, into GPT-4o. We ask GPT-4o to rank all visible objects based on their helpfulness in finding the navigation target. For each navigation target, we collect three such rankings corresponding to three types of descriptions.

### A.3.2 TRAINING PROCESS

We fine-tune our model based on the LLaVA-1.5-7B checkpoint(Liu et al., 2023) using the collected training dataset for 5 epochs with a learning rate of 4e-6 and a batch size of 1. We use the AdamW optimizer with no weight decay. During training, DeepSpeed ZeRO-2 and LORA (Hu et al., 2021) are used to save GPU memory and accelerate training. FP16 is enabled to balance speed and precision. We train our model with $6\times24$ Tesla V100 GPUs, and the fine-tuning process is completed within 6 hours.

We use the default CLIP vision encoder of LLaVA to encode all memory snapshots, frontier snapshots, egocentric views and image navigation targets. And the encoded vision features are further compressed to $12 \times 12$ (for image targets and egocentric views) and $3 \times 3$ (for memory snapshots and frontier snapshots) tokens in the training prompt.

During fine-tuning, we simultaneously optimize the model for exploration task and prefiltering task with cross-entropy loss. The loss weights for exploration and prefiltering are set to 1 and 0.3, respectively. The training goal of exploration is to correctly predict the ground truth choice of memory snapshot or frontier at each step. The training goal of prefiltering is to select the top 10 helpful objects that have been observed, based on the ground truth we collected earlier.

### A.4 ABLATION STUDY

We mainly evaluate on the number of egocentric observations at each step ($N$), the maximum distance an object should be included in the memory snapshot ($max\_dist$), and the number of prefiltered classes ($K$).

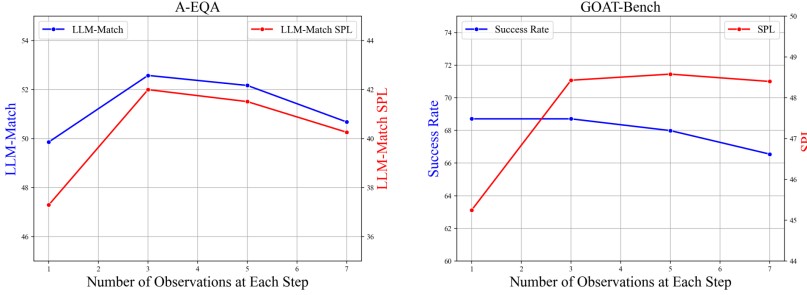

Figure 6: Ablation on the number of observation each step ($N$) for A-EQA and GOAT-Bench.

In Figure 6, we present the evaluation metrics for different choices of $N$ on both A-EQA and GOAT-Bench. We can observe that increasing the number of observations does not necessarily lead to better performance. This is mainly because the additional views often provide repeated and redundant information. Furthermore, as the number of frame candidates increases, a cluster of objects that would originally be assigned to one memory snapshots may instead be assigned to separate memory snapshots, resulting in confusion. Based on the results, we choose $N = 3$ for both datasets.

In Figure 7, we present the evaluation metrics for different choices of $max\_dist$ on both A-EQA and GOAT-Bench, where we observe different tendencies across the two benchmarks. Evaluation metrics on GOAT-Bench generally improve with an increase in $max\_dist$, while metrics on A-EQA decline. This is because, under normal circumstances, a memory snapshot should only represent objects within a local area. Objects in more distant regions should either remain in unexplored areas or be captured by another memory snapshot that is closer to them. A large $max\_dist$ imposes a looser distance restriction, which can introduce disorder. However, in the navigation task of GOAT-Bench, the earlier the target object is added to the scene graph as a choice for the VLM, the faster

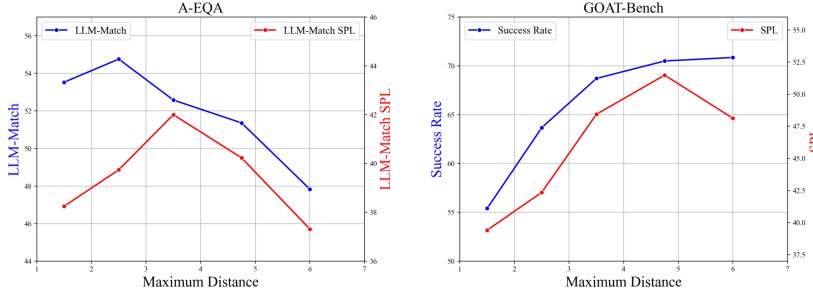

Figure 7: Ablation on the maximum distance for including an object to the scene graph ($max\_dist$) for A-EQA and GOAT-Bench.

the VLM can select it as the direct navigation target, resulting in faster arrival at the target objects. Balancing both accuracy and efficiency across the two benchmarks, we choose $max\_dist$ to be 3.5 meters.

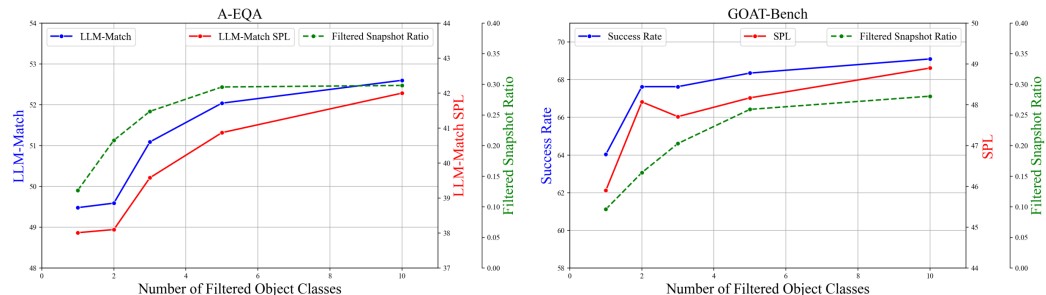

Figure 8: Ablation on the number of prefiltered classes ($K$) for A-EQA and GOAT-Bench.

In Figure 8, we present the evaluation metrics for different choices of $K$ on both A-EQA and GOAT-Bench. In addition to the metrics introduced in the experiment sections, we include the average ratio of the number of remaining memory snapshots after prefiltering to the total number of memory snapshots as a measure of the effectiveness and intensity of prefiltering. The results on both benchmarks align with our intuition: allowing more prefiltered classes leads to better performance. Moreover, even when $K = 10$, on average only 3.26 and 4.66 memory snapshots are left after prefiltering for A-EQA and GOAT-Bench respectively, accounting for 29.8% and 28.1% of the total memory snapshots, and 8.2% and 5.1% of the total frame candidates. These statistics demonstrate the effectiveness of prefiltering as a memory retrieval mechanism, as well as SnapMem's compactness as a scene representation. Furthermore, we observe that the overall performance does not drop significantly when $K$ is small, highlighting the robustness of our framework.

## A.5 COMPLETE PROMPTS FOR VLMS

We present the full prompt for prefiltering in Figure 9, the prompt for embodied question answering (A-EQA dataset) in Figure 10, and the prompt for navigation (GOAT-Bench dataset) in Figure 11.

918
919
920
921
922
923
924
925
926
927
928
929
930
931
932
933
934
935
936
937
938
939
940
941
942
943
944
945
946
947
948
949
950
951
952
953
954
955
956
957
958
959
960
961

---

**System Prompt:**

You are an AI agent in a 3D indoor scene.

**Content Prompt:**

Your goal is to answer questions about the scene through exploration.
To efficiently solve the problem, you should first rank objects in the scene based on their importance. These are the rules for the task.
1. Read through the whole object list.
2. Rank objects in the list based on how well they can help your exploration given the question.
3. Reprint the name of all objects that may help your exploration given the question.
4. Do not print any object not included in the list or include any additional information in your response.

Here is an example of selecting helpful objects:
Question: What can I use to watch my favorite shows and movies?
Following is a list of objects that you can choose, each object one line:
painting
speaker
box
cabinet
lamp
tv
book rack
sofa
oven
bed
curtain
Answer:
tv
speaker
sofa
bed

Following is the concrete content of the task and you should retrieve helpful objects in order:
Question: {question}
Following is a list of objects that you can choose, each object one line:
{class_0}
{class_1}
...
Answer:

---

Figure 9: Prompt for prefiltering. The placeholders {question} and {class_$i$} are replaced by the question and all existing classes in the scene graph, respectively.

962
963
964
965
966
967
968
969
970
971

---

**System Prompt:**

Task: You are an agent in an indoor scene tasked with answering questions by observing the surroundings and exploring the environment. To answer the question, you are required to choose either a Snapshot as the answer or a Frontier to further explore.

Definitions:
Snapshot: A focused observation of several objects. Choosing a Snapshot means that this snapshot image contains enough information for you to answer the question. If you choose a Snapshot, you need to directly give an answer to the question. If you don't have enough information to give an answer, then don't choose a Snapshot.
Frontier: An observation of an unexplored region that could potentially lead to new information for answering the question. Selecting a frontier means that you will further explore that direction. If you choose a Frontier, you need to explain why you would like to choose that direction to explore.

**Content Prompt:**

Question: {question}
Select the Frontier/Snapshot that would help find the answer of the question.

The following is the egocentric view of the agent in forward direction: [img]

The followings are all the snapshots that you can choose (followed with contained object classes).
Please note that the contained classes may not be accurate (wrong classes/missing classes) due to the limitation of the object detection model. So you still need to utilize the images to make decisions.
Snapshot 0 [img] {class_0}, {class_1}, ...
Snapshot 1 [img] {class_0}, {class_1}, ...
...

The followings are all the Frontiers that you can explore:
Frontier 0 [img]
Frontier 1 [img]
...

Please provide your answer in the following format: "Snapshot i\n[Answer]" or "Frontier i\n[Reason]", where i is the index of the snapshot or frontier you choose. For example, if you choose the first snapshot, you can return "Snapshot 0\nThe fruit bowl is on the kitchen counter.". If you choose the second frontier, you can return "Frontier 1\nI see a door that may lead to the living room.".
Note that if you choose a snapshot to answer the question, (1) you should give a direct answer that can be understood by others. Don't mention words like "snapshot", "on the left of the image", etc; (2) you can also utilize other snapshots, frontiers and egocentric views to gather more information, but you should always choose one most relevant snapshot to answer the question.

---

Figure 10: Prompt for embodied question answering. The placeholders {question} and {class_$i$} are replaced by the question and the object classes contained in the corresponding memory snapshots, respectively. [img] are replaced by the egocentric views, memory snapshots or frontier snapshots.

**System Prompt:**

Task: You are an agent in an indoor scene that is able to observe the surroundings and explore the environment. You are tasked with indoor navigation, and you are required to choose either a Snapshot or a Frontier image to explore and find the target object required in the question.

Definitions:
Snapshot: A focused observation of several objects. It contains a full image of the cluster of objects, and separate image crops of each object. Choosing a snapshot means that the object asked in the question is within the cluster of objects that the snapshot represents, and you will choose that object as the final answer of the question. Therefore, if you choose a snapshot, you should also choose the object in the snapshot that you think is the answer to the question.
Frontier: An unexplored region that could potentially lead to new information for answering the question. Selecting a frontier means that you will further explore that direction.

**Content Prompt:**

Question: {question}
Select the Frontier/Snapshot that would help find the answer of the question.

The following is the egocentric view of the agent in forward direction: [img]

The followings are all the snapshots that you can choose. Following each snapshot image are the class name and image crop of each object contained in the snapshot. Please note that the class name may not be accurate due to the limitation of the object detection model. So you still need to utilize the images to make the decision.
Snapshot 0 [img] Object 0: {class_0} [img_crop_0], Object 1: {class_1} [img_crop_1] ...
Snapshot 1 [img] Object 0: {class_0} [img_crop_0], Object 1: {class_1} [img_crop_1] ...
...

The followings are all the Frontiers that you can explore:
Frontier 0 [img]
Frontier 1 [img]
...

Please provide your answer in the following format: "Snapshot i, Object j" or "Frontier i", where i, j are the index of the snapshot or frontier you choose. For example, if you choose the fridge in the first snapshot, please return "Snapshot 0, Object 2", where 2 is the index of the fridge in that snapshot. You can explain the reason for your choice, but put it in a new line after the choice.

Figure 11: Prompt for GOAT-Bench dataset. The placeholders {question} and {class_$i$} are replaced by the question and the object classes contained in the corresponding memory snapshots, respectively. [img] are replaced by the egocentric views, memory snapshots or frontier snapshots, and [img_crop_$i$] are replaced by the corresponding object crops, which are directly cropped from the memory snapshots based on the detection bounding boxes.

