# OpenReview forum: "SnapMem: Snapshot-based 3D Scene Memory for Embodied  Exploration and Reasoning"
_ICLR.cc/2025/Conference — ICLR 2025 Conference Withdrawn Submission_

### Official Review · Reviewer_bvwf · 2024-11-02

**Soundness:** 2
**Presentation:** 2
**Contribution:** 1
**Rating:** 3
**Confidence:** 4

**Summary:**

This work proposes SnapMem which uses a set of posed RGBD images that best covers all objects in the scene as the scene memory. By querying Vision Language Models (VLMs) with these snapshot memories, this work develops a rule-based framework to achieve frontier exploration. It demonstrates advantageous empirical results on three embodied question-answering (EQA) benchmarks over traditional scene representations.

**Strengths:**

1) The idea of preserving key images as scene memory to allow VLMs to do better at embodied exploration and reasoning is reasonably justified.
2) The experiment results on three tasks consistently demonstrate good performance supporting the strength of the proposed representation over the baselines.
3) Detailed prompts are provided in the appendix for reproducibility.

**Weaknesses:**

W 1.  Fundamentally, the reviewer has concerns about the problem setup. Replying on knowing the poses and coordinates of images and objects has great limitations when deploying real embodied agents in 3D scenes. It is a really strong piece of information especially for unknown environments. If you already have images with known poses and objects with known coordinates, it is not far from having a 3D scene graph with rather complete global information about the scene, which weakens the motivation of using snapshot images as alternative scene representation, as it depends on knowing the same amount of 3D information. Or is the proposed SnapMem only meant to be a better interface for VLMs?

W 1.1 Since the SnapMem is built on object co-visibility, which is almost in 2D, it is probably more meaningful to build your pipeline in 2D, when poses and coordinates are not available or not reliable, and 3D scene graphs or other 3D representations are difficult to obtain.

W 2. Concerns about the method. Relying only on objects when choosing the snapshot is also a potential issue. What if there are areas in an environment that do not contain any objects ( or objects that can be detected by the model), they will not likely be captured by SnapMem though they can be captured by 3D scene representations, so are they not important to the scene and to an embodied AI robot at all?

W 2.1 The SnapMem system seems very rule-based and has a bunch of predefined rules and hyperparameters (e.g. the value of the *max_dist*, K in Prefiltering, etc.). How do the rules and hyperparameters generalize to different environments? I see that these values are tuned for the three tasks.

W 3. Missing related works. The paper includes related works on 3D scene representations but is missing discussions on 2D scene representations. The proposed SnapMem is a set of images that falls in 2D. Literature in Topological Mapping also builds scene representations in 2D [1, 2, 3, 4]. For example, in [2] and [4], their representation also contains images and objects in 2D. Discussions on why SnapMem is a better representation for exploration and embodied QA is appreciated.


---
[1] Blochliger, Fabian, et al. "Topomap: Topological mapping and navigation based on visual slam maps." 2018 IEEE International Conference on Robotics and Automation (ICRA). IEEE, 2018.

[2] Garg, Sourav, et al. "Robohop: Segment-based topological map representation for open-world visual navigation." ICRA (2024).

[3] Chaplot, Devendra Singh, et al. "Neural topological slam for visual navigation." CVPR. 2020.

[4] Kim, Nuri, et al. "Topological semantic graph memory for image-goal navigation." Conference on Robot Learning. PMLR, 2023.

**Questions:**

1. to clarify: what does the "same observation" mean on line 226? According to line 20 in Algo 1, does that mean the same image and the same object set?

2. to clarify: at line 220, what is the "confidence" referring to? I don't see it defined or mentioned anywhere else in the paper.

More questions are mentioned in the Weaknesses section.

---

### Official Review · Reviewer_TSx9 · 2024-11-03

**Soundness:** 3
**Presentation:** 3
**Contribution:** 3
**Rating:** 6
**Confidence:** 4

**Summary:**

This paper introduces SnapMem, a snap-shot-based 3D representation for memory-efficient embodied reasoning and action. SnapMem focuses on providing an efficient representation of indoor scenes by utilizing images with semantic information, leading to lifelong autonomous agents. Extensive experiments on 3 different datasets demonstrate the effectiveness of SnapMem.

**Strengths:**

1. This paper proposes SnapMem that achieves better performance across 3 different benchmarks, illustrating the generalization ability of the proposed method.
2. SnapMem is memory efficient and a promising way to achieve lifelong autonomy, which is clearly and precisely stated in the paper.
3. The paper writing is clear and easy to follow.

**Weaknesses:**

1. Despite being stated in the paper that SnapMem outperforms traditional scene representations, it is not clear from the experiments whether SnapMem is able to outperform 3D representations like point cloud.
2. The paper does not analyze the failure cases for the method. For example, it is more helpful if metrics corresponding to different question types are provided.
3. In addition to 2, since SnapMem can be incorporated into any VLM, it is crucial for readers to understand what the performance bottleneck is, is it the pre-filtering of images? Or is it VLM's ability to reason?

**Questions:**

1. For the EM-EQA task, is it possible to report the performance of an LLM that takes 3D point clouds as input, for example, 3D-LLM[1] or LEO[2]?


[1] Hong, Yining, et al. "3d-llm: Injecting the 3d world into large language models." Advances in Neural Information Processing Systems 36 (2023): 20482-20494.

[2] Huang, Jiangyong, et al. "An embodied generalist agent in 3d world." arXiv preprint arXiv:2311.12871 (2023).

---

### Official Review · Reviewer_ptdH · 2024-11-03

**Soundness:** 3
**Presentation:** 3
**Contribution:** 2
**Rating:** 3
**Confidence:** 4

**Summary:**

The paper introduces "SnapMem," a snapshot-based memory system designed for 3D scene representation to support embodied agents in complex exploration and reasoning tasks. Unlike object-centric representations, SnapMem uses clusters of co-visible objects captured in snapshot images to convey richer spatial and semantic information. These memory snapshots, along with unexplored "frontier snapshots," guide agents in frontier-based exploration and help construct an incremental memory structure. SnapMem also includes an efficient retrieval system to filter relevant snapshots for reasoning tasks. Experimental results across three benchmarks show that SnapMem improves both reasoning accuracy and exploration efficiency in 3D environments.

**Strengths:**

1. The memory retrieval technique (prefiltering) reduces the computational load by only retrieving relevant snapshots. While the improvement may not be groundbreaking, it contributes to the efficiency of the memory system.
2. The paper evaluates SnapMem across multiple benchmarks, showing that it can be adapted to different embodied AI tasks, such as question answering and navigation. This suggests potential flexibility for diverse applications.

**Weaknesses:**

1. The contribution in the proposed methods is mostly incremental. While not maintaining a graph, the object-image relations and frontier idea are all common in related spatial memory and scene representation work. The authors need to be clearer on the key differences and contributions of the proposed methods. The prefiltering and exploration part is mostly based on prompting VLMs; the authors need to explain and show the motivation and why other methods do not work.

2. The clustering method relies on a basic rule-based approach using co-visibility in snapshots. This method lacks sophistication, as it does not leverage advanced clustering or segmentation techniques that could more accurately represent spatial relationships within the scene. The authors should explain why more sophisticated learning methods do not work for this task.

3. The prefiltering can be a good approach, but the authors do not provide any information on the number of images before and after prefiltering. Neither do the experiments show the difference in performance with and without prefiltering.

4. The evaluation relies heavily on standard benchmark scores without providing an in-depth analysis of how SnapMem handles specific scene complexities (e.g., cluttered scenes, varying lighting conditions). The authors are suggested to provide more in-depth analysis either by introducing new metrics or discussing more aspects of the current results.

5. GPT-4o is the only VLM tested. This is a closed model and hinders reusability and reproducibility by the community. The authors are suggested to also provide performance on other open-source VLMs.

6. Typo: Line 219, I*> seems to be missing a symbol.

**Questions:**

If the authors are able to address the concerns listed above, I am happy to raise my rating.

---

### Official Review · Reviewer_Qg8Z · 2024-11-04

**Soundness:** 3
**Presentation:** 2
**Contribution:** 3
**Rating:** 5
**Confidence:** 5

**Summary:**

This paper proposes a novel image snapshot based scene memory structure for navigation task, SnapMem. SnapMem consists of clusters of images that are constructed considering co-visible objects during the exploration. Since SnapMem captures not only observed object relations but also their surrounding images, it can store more robust information and can also be extended to represent frontier snapshots. Through dynamic memory aggregation and memory retrieval with prefiltering, SnapMem enables agents to actively expand their knowledge with efficiency and supports lifelong learning in 3d environment.

**Strengths:**

The proposed idea of utilizing visual information in addition to the object relation is sound and reasonable. VLM is appropriately used to utilize this visual information. The idea of considering frontier based exploration seems reasonable and effective for navigation tasks. The experimental results show the successful results of the proposed method in LLM-Match and efficiency.

**Weaknesses:**

Although the proposed idea is very reasonable and the experimental results seem promising, the manuscript does not fully deliver explanations of the proposed method.
For example, in algorithm 1, notations like ||O||, F(I) = ||O_I|| are not clear which makes hard to understand the algorithm. It seems the pipeline of the algorithm is based on ConceptGraph but there is no explanation of its detail which also makes hard to understand the algorithm solely with this paper.
It seems frontier snapshots are included in the same SnapMem structure with other memories with the same configuration, but the method to integrate those two different types memories is not clear. Frontier snapshot does not seem to include object detection but the clustering algorithm is based on the detected object so it would be better to describe how to integrate them more clearly.
The process of the proposed method in navigation running time (test situation) is not clear. Although Figure 2 shows the memory aggregation process of SnapMem, the manuscript lacks the navigation pipeline including that process. It would be better to include some more details for that such as the frequency of capturing images, the frequency of running algorithm, the average memory size of SnapMem per one navigation episode, etc.
For the experimental results, it would be better to add an efficiency column in table 1 that represents how many images (in memory) are use. In table 2., the column Avg. frames is not clear. I guess that it is not an average number of frames in the whole SnapMem of an episode. It would be more clear to claim the efficiency of the proposed algorithm. Also, there is no description of the navigation setting such as action space (discrete or continuous)
There are many missing citations in the manuscript (e.g, 3.1 ConceptGraph). Also, some notations should be reviewd (e.g., 3.2.2 I_t = I_(t-1) + I)

**Questions:**

- Some clarification for the notations in algorithm: ||O||, ||O_I||
- How to integrate snapshot memory with object detection and frontier snapshot together?
- What is the frequency of capturing observation images in the navigation? I guess it is related to the "average frame" in the experiment.
- What is the frequency of running the SnapMem update algorithm and inference through VLM, related to the number of action steps?
- What is the setting of the navigation agent? (action space)
- What is the navigation policy for the experiment?
- Is there any usage of spatial information of snapshot memory? It seems positional information is not used for constructing the SnapMem but the Figure 1. shows graph-like structure with edge-like connection between clusters
- Is there any guideline for the VLM prompt? For example, constraints on object names or pool of question types.

---

### Note · Authors · 2024-11-14

I have read and agree with the venue's withdrawal policy on behalf of myself and my co-authors.